# The Experiences of Informal Carers during the COVID-19 Pandemic: A Qualitative Systematic Review

**DOI:** 10.3390/ijerph192013455

**Published:** 2022-10-18

**Authors:** Cara Bailey, Ping Guo, John MacArtney, Anne Finucane, Susan Swan, Richard Meade, Ellie Wagstaff

**Affiliations:** 1School of Nursing and Midwifery, Institute of Clinical Sciences, College of Medical and Dental Sciences, University of Birmingham, Birmingham B15 2TT, UK; 2Warwick Medical School, University of Warwick, Coventry CV4 7AL, UK; 3Clinical Psychology, School of Health in Social Science, University of Edinburgh, Edinburgh EH8 9YL, UK; 4Marie Curie Hospice Edinburgh, Edinburgh EH10 7DR, UK; 5Maggie’s Glasgow, Gartnavel Hospital, 1053 Great Western Road, Glasgow G12 OYN, UK; 6Carers UK, 20, Dover Street, London SE1 4LXT, UK; 7Marie Curie UK, Edinburgh EH10 7DR, UK

**Keywords:** carer, COVID, pandemic, informal care, family caregiver

## Abstract

Objectives: To identify, critically appraise and synthesise the qualitative literature on the experiences of informal carers of people with long-term conditions during the COVID-19 pandemic. Design: A qualitative systematic literature review. Data Sources: Eight electronic databases were systematically searched (Medline, Embase, CINAHL, PubMed, PsychINFO, Web of Science, Nursing and Allied Health and ASSIA) along with Google Scholar and handsearching via secondary sources. Study selection: Eligible studies had to include the experiences of informal carers (adults who are 65 or older), use a qualitative methodology and had to be written in English. Data extraction and synthesis: Retrieved papers were quality assessed using the Critical Appraisal Skills Programme qualitative checklist and ranked for quality. Thematic analysis was used to synthesise the findings. Results: Fourteen studies were included, all from medical or nursing journals (*n* = 5 specifically gerontology). Four main themes were identified: (i) fear, (ii) uncertainty, (iii) burden and (iv) staying connected. Caregiving demands have increased for carers during the pandemic, as well as negative emotions such as fear and uncertainty. At the same time, less social support has been available, leading to concerns about carers’ wellbeing and ability to cope. Conclusion: Carers’ needs have been exacerbated by the COVID-19 pandemic. Greater practical and emotional support is needed for carers from both formal services and community sources that considers their changing needs and offers educational and emotional support for long-term wellbeing. Strengths and Limitations: (1) This is the first systematic review to explore in depth the experiences of informal carers caring for people with a range of long-term conditions and from an international context. (2) The review includes an analysis of the quality of the studies, as well as a study of their relative contributions. (3) Further research is needed to explore the physical, emotional and financial impact of the pandemic for bereaved carers which is not captured in this review due to the lack of empirical data available at the time of review.

## 1. Introduction

The ongoing coronavirus pandemic, declared by the World Health Organisation (WHO) on the 11th of March 2020 [1] (WHO, 2020), has had a devastating impact globally, with physical, psychological, social, and economic implications for society. An informal carer—defined as a person who provides unpaid practical care to another person such as a relative, friend or neighbour [2]—is a crucial part of many healthcare systems. For example, in the United Kingdom (UK), it was estimated that there were 9.1 million unpaid carers prior to the pandemic, but that this has grown substantially to 13.6 million since the start of the outbreak [3]. It is estimated that the value of care provided by unpaid carers in the UK during the pandemic was GBP 530 million. per day [4]. Such figures demonstrate the number of people caring for older, disabled or seriously ill relatives and friends within communities, often with no formal training or preparation.

Public health measures to reduce the spread of the virus, such as social distancing and in some cases closure of in-person nursing, social care and support services, meant that the caregiving burden increased quickly and intensely. During the pandemic, many relatives, friends and neighbours became ‘carers’ almost overnight, at a time where support (both formal and informal) was restricted. The Global Carer Well-being Index, a survey of 12 countries, reported that time needed for caregiving has increased rapidly since the start of the pandemic, with 20% of the adult population stepping up to a caregiving role for the first time [5]. Research conducted about population ageing, indicated that informal caregiving is associated with poorer mental health and increased social isolation compared to the general population [6]. A global survey found that 61% of informal carers reported that their mental and emotional health worsened since the beginning of the pandemic [5]. Now more than ever, attention and support are needed to protect the wellbeing of carers and consequently that of their care recipients.

The systematic review reported here aimed to retrieve, critically appraise, and synthesise the qualitative evidence on the experiences of informal carers during the COVID-19 pandemic to identify key themes within this population. Understanding in depth the issues and challenges that carers faced during and in the aftermath of this public health crisis is essential for informing policy and improving services to better support carers and their changing needs in the future.

## 2. Methods

### 2.1. Design

The systematic review follows the Preferred Reported Items for Systematic Reviews and Meta-Analyses (PRISMA) statement and guidelines [7]. The systematic approach ensured the evidence was reviewed rigorously and reduced the risk of researcher bias, which could otherwise compromise the credibility of the results.

### 2.2. Search Strategy

Eight databases were searched including MEDLINE, Embase, CINAHL, PubMED, PsychINFO, Web of Science Core Collection, Nursing and Allied Health, and Applied Social Sciences Index and Abstract (ASSIA) between April 2020 and January 2022, considered to retrieve most optimal coverage for the topic [8]. Papers from Google Scholar were retrieved by ‘Publish or Perish’ [9]; the first 100 results of the search were retrieved for the screening process. Manual searching through references of relevant papers was conducted to identify additional papers [10]. The review was not limited by country to allow a comprehensive investigation of the evidence.

The search terms (Table 1) were generated from the use of the Population, Exposure, Outcome (PEO) framework for qualitative research questions [11]. We recognise the variation in terms used for informal carers, unpaid carers, family carers and caregivers and whilst we include all variations in our searches, for consistency in reporting we use the term informal carers. Synonyms and medical subject headings (MeSH) were also used in the relevant databases to expand the search results with Boolean operators and truncation.

Database search results were limited to English language (due to practical reasons of no access to translation services).

Inclusion and exclusion criteria were used to identify relevant papers to be included in the review [12] to reduce bias during the screening process [13]. To meet the inclusion criteria, the papers had to be empirical studies with qualitative design that were peer-reviewed. Studies investigating carers and care recipients under the age of 18 were excluded, as these groups would likely have significantly different experiences and implications. Papers that investigated the experience of caregiving for people who contracted coronavirus and the experience of paid carers were excluded as this review was primarily concerned with the experiences of informal carers of people with advanced illness or long-term conditions.

### 2.3. Quality Appraisal

The Critical Appraisal Skills Programme (CASP) checklist tool (CASP, 2019) for qualitative research [14] was used to systematically assess the methodological quality of each study to ensure a good measure of transparency of research standards and reporting practices [15]. The appraisal conducted by RT and CB assessed the quality, relevance, and usefulness of papers, and examined any potential biases that may have affected the results of the study, reducing its credibility [16].

### 2.4. Data Extraction

A data-extraction table was created to summarise the key characteristics, findings, and limitations of the studies in a manageable form, to enable analysis. The authors, year, aim, study design, sample population and size, country, data analysis, main themes and limitations of each study were described.

### 2.5. Data Analysis

The main themes as described in the papers were extracted and synthesised using Braun and Clarke’s method of thematic analysis [17], involving a six-step process of (1) familiarisation; (2) coding; (3) generating themes; (4) reviewing themes; (5) defining and naming themes and (6) writing up. Firstly, the data in the papers were read multiple times to enable familiarisation (1), then codes were applied to transcribed sections of text to establish the general concepts identified in each study by the first reviewers CB and RT using the extracts, quotes and key themes in the papers (2). Patterns were identified within and between the codes which were then used to develop overarching themes within the data by RT and CB (3). A mind map was created to ensure that the themes were appropriately generated from the codes and collaborated with the wider research team (4) (See Appendix A). Themes were then defined and the data used to create a synthesis using direct quotations from the data as evidence (5) and written up (6).

### 2.6. Patient and Public Involvement

No patients involved.

## 3. Results

### 3.1. Search Outcome

A total of 1041 papers were retrieved from the database searches, with a further 100 retrieved from Google Scholar and 3 papers retrieved through manual searches (268 duplicates were removed). The titles and abstracts were screened (by CB and RT) according to the inclusion and exclusion criteria; 58 records were sought for retrieval of which 44 of these were discarded according to the exclusion criteria. No studies were excluded from the review based on their quality. A total of 14 papers were suitable for the review (Table 2).

### 3.2. Critical Appraisal

Papers were assessed on each aspect of the Critical Appraisal Skills Programme qualitative checklist and given a rating of ‘poor’, ‘fair’ or ‘good’ (Table 2). All papers used a qualitative research design or had a qualitative element, appropriate for the aim of exploring carers’ lived experiences. Most studies used semi-structured interviews and were conducted on the telephone or web-based platforms, apart from a survey with open responses [26,29]. Convenience or purposive sampling was most used with just one study randomly selecting participants as part of a larger trial [29]. Purposive sampling can be useful to ensure inclusion of different socioeconomic backgrounds, severity of illness and to ensure an equal distribution of demographics such as age and gender [25]. However, the majority of the sample reviewed were women which is unsurprising, as most informal carers are women [32], therefore these samples may accurately reflect the gender distribution of carers. A common shortcoming in all the studies was that none of the researchers critically analysed the relationship between the researchers and the participants. However, one study [19] involved people with dementia and informal carers of people with dementia in the formation of their interview questions and in the interpretation of findings to ensure the research reflected their lived experiences, increasing the validity of their results. All the papers were deemed to be of fair (+) or good (++) quality and suitable for inclusion in the thematic analysis.

### 3.3. Study Characteristics

The 14 studies reviewed collectively report the experiences of 476 carers of people with long-term conditions (summarised in Table 3). Six papers reported the experiences of carers of people living with dementia [19,22,24,25,27,31]. A further two papers were related to mental health conditions [28,30] and another examined carers for people with neuro-palliative care conditions (mainly Parkinson’s and Alzheimer’s disease) [29]. One study [18] focused on carers of people with end-stage renal disease. Two studies focused on carers of people with cancer [23,25], one focused on caregiving of older adults and adults with learning disabilities [20], and another on the experiences of carers of stroke survivors [21].

Research reported in the included papers was conducted in the UK [19,27,28,30,31], US [20,26,29], Hong Kong [21], India [25], Singapore [23], Italy [24], Portugal [18], and Poland [22]. They include a range of cultures (different ethnic and gendered divisions of roles of family and friends in caring) and different healthcare systems (national, insurance and mixed) with related normative expectations (from individual responsibility to state welfare). It should also capture the differing rates of infection, deaths and impacts on healthcare services (including those that pursued zero-COVID-19 policies [21,23] to provide an international perspective of the impact of on informal carers.

#### Thematic Findings

Four overarching themes (and sub-themes) of informal carers’ experiences during the pandemic were identified: (1) fear (of viral spread and wider sense of danger); (2) uncertainty (restriction rules and new systems of care processes); (3) burden (protective measures, closure of support services and increased expectations; and (4) staying connected (loneliness and isolation, finding alternatives). Appendix A illustrates how the data make up the key themes and then discussed using examples from the data.


**Fear—“*We are always in apprehension*”**
[24]

Fear was evident in the majority of studies [18,20,21,23,26,27,30,31] with the main fear held by carers being that their care recipient would contract COVID-19 and could die as a result, due to being vulnerable [1,23,24]. Carers were also fearful of the wider sense of threat and danger in society and the implications it may have on the care the care recipient would receive.

Carers reported they felt afraid of infecting their family members and were anxious even when taking all the necessary safety precautions [24,25,26,27,30].

“*Let’s say that right now it is even more difficult because we are always feeling the panic of transmitting her the virus. As soon as we see that she has a temperature or she is more agitated, then we immediately think, ‘It must be this [COVID-19].’ We are always in apprehension*” (Carer [24])

Knowledge of asymptomatic transmission and carrying out public errands such as grocery shopping exacerbated this fear [18,22,24,25,27,31]. Some identified a common fear among carers that their care recipient would be hospitalised and they would not be able to visit them due to ‘no visitation’ hospital rules [18,24,25,31].

“*I didn’t want him to go to hospital*” (Carer [31])

Similarly, others changed their place of care preferences due to the fear of not being able to visit.

‘*The caregiver is not following through with placing him in a nursing home because she is afraid she will never see him again*’ (From medical notes [29])

Carers seemed less concerned about their own vulnerability to the virus [18], as they prioritised their care recipient’s health over their own, even to the point of causing negative impacts on their own quality of life. Observing people breaking lockdown rules in the media and in public increased feelings of stress and frustration as carers felt they were being put at risk by those being irresponsible [19,23,24]. Carers also felt fearful when observing health and social care staff failing to adhere to protective measures such as handwashing and use of PPE, with some cancelling paid care in fear of them transmitting the virus [19,20,24]. Even when lockdown restrictions were eased, carers were still fearful of transmission and this was still impacting on their mental state [19,21,26].


**Uncertainty—“*We have no control over it, there seems to be no end to it*”**
[19]

Carers experienced constant uncertainty during the pandemic [18,19,22,23,26,27,30]. Uncertainties related to the nature of the changing rules and new systems of care being implemented with limited communication of information.

Initially, little was understood about the virus [23], ‘rules’ were changing rapidly, and advice was often unclear [31].

“*The government advice has been so ambiguous and changing*” (Carer [31])

Lockdowns in various geographical regions caused uncertainty about the trajectory of the pandemic and therefore the future of support services for care recipients was a worry for carers [22,23,24,28,29]. The uncertainty resulted in a loss of hope for some of the carers who could not see an end to the restrictions which culminated in a further sense of ‘vulnerability’.

“*Everybody’s vulnerable and with the virus we’re even more vulnerable*” (Carer [29])

Carers worried about the implications of the pandemic on their care recipient’s treatment, especially in regard to cancer therapy [23,26], with treatments being postponed and the implications regarding treatment if the care recipient caught COVID-19. One carer said that they had to be “*very aggressive*” to get information from health care professionals about their parent’s care in relation to the pandemic [26]. Lack of clear information was evident in other studies [18], and the remote nature of ‘support’ negatively impacted on the opportunities for shared decision making which was important for complex cases especially in mental health [28]. Carers, many of whom had no training or preparation for their role as a ‘carer’, reported they wanted more information from healthcare professionals about ‘how to care’ so that they could feel better prepared for this situation [18].

“*I think there is a lack of closer care, a greater explanation of what is going on and what could happen*” (Carer [18])

Economic uncertainty and stress were also expressed by carers in several studies [18,24,25,30] that reported uncertainty related to work status and the impact of the pandemic on the economy and personal financial risks in relation to carer benefits.

“*I was really frightened all the time of losing my benefits*” (Carer [30])

Carers reflected on their future at the time of uncertainty [22,30] and particularly worried about who would look after their care recipient if they fell ill [19,20,26], even in countries where broad sanctions were imposed and welfare support and benefits were offered


**Burden—“*It’s ground hog day*”**
[27]

Burden was evident in the majority of studies [18,20,21,23,25,26,27,28,29] and appeared to be directly exacerbated by the pandemic-related restrictions in place. In particular, the responsibility of protective measures directly impacted on day-to-day activities; the closure of necessary services that enhanced quality of life and offered respite and socialisation; and the increased expectations as a result of being a sole carer. Some also reported that symptoms were exacerbated amongst the people being cared for during the pandemic as a result of the drastic change in routine and restrictions on activity [25,29].

“*Patient became depressed due to covid which resulted in self-harm*” (Clinical note extract) [29]

A significant additional responsibility of informal carers in all studies was the protective measures taken to prevent their care recipient from being infected with the virus, such as the use of protective personal equipment, disinfecting items, and social distancing. Reminding the care recipient about protective measures became a demanding task for many participants, as some care recipients did not understand the importance of these measures, which was notably more challenging for carers of people with dementia [19,22,24,27,31].

“*[The care recipient asks] ‘Why are you wearing that mask?’ and I reply to her, ‘Because, Mum, there is a virus, and I could also infect you, so it is better to have my mask on’, and she asks me this every time, even five minutes later*” (Carer [24])

“*I have to say, mom don’t be sad or mad, this is protect you and us*” (Carer [18])

The change in routine caused confusion for the person with dementia and carers encountered great difficulty in getting care recipients to follow the measures especially in terms of socially distancing [22,31].

“*We wouldn’t be able to socially distance because we can’t make people living with dementia do it*” (Carer [22])

Some care recipients became ‘irritable’ when prompted to wear a mask [24] responding with aggression. Some carers reported feeling embarrassed due to their care recipients’ difficulty in following the rules in public and felt shamed by others as a result [19,25] and that they wanted more awareness in society of the difficulties of living with dementia during the pandemic [19,25].

“*We’re getting cards…to say that he has a disability, but it’s also a case of some people still won’t understand that, so I’m anxious as to how that’s going to go*” (Carer [19])

In response to behavioural issues, some carers adopted a more relaxed approach to the protective measures inside their home in order to cause less distress to their care recipient. Some said they stopped taking their relatives out in public due to the difficulty of getting their family members to comply with public safety measures [19,24,25,27].

As support services closed, the burden increased [18,20,21,22,24,25]. This was particularly a source of stress for carers of people with dementia as care recipients were no longer receiving social stimulation [22,25].

“*What I felt when they closed the day care facility? It was stress and it still is. We had to organise something that worked well again. I didn’t know when it was going to end*” (Carer [22])

“*It’s a groundhog day. Every day, every single day she does not understand why she cannot go outside for a walk, every single day we go through the same thing*” (Carer [27])

Some services were offered online, which was appreciated by carers, however most reported that it was not as beneficial as in-person services [20,22,28,29,30]. Carers were distressed as they observed a deterioration in their care recipient’s symptoms [29], which threatened the relationships between carer and person [21] and impacted on their own wellbeing [24].

“*I have seen a considerable worsening, let’s say; he wants me always close to him and obviously this means that I don’t have any time for me*” (Carer [24])

As rehabilitation centres [21], day care services [19,29] and social care support networks stopped being face to face [27], people missed social connection and the benefits of the therapeutic setting [28] which had a detrimental effect on symptoms that increased for stroke [21], mental health [28] and neuro-palliative care patients [29]. In the absence of formal support structures, carers tried to take on the role of rehabilitator for the care recipient, which was physically demanding and stressful as they did not feel competent or confident enough to perform the role well [21,25,29].

Carers wanted more support regarding caregiving in a variety of areas. Participants expressed desire to learn more skills related to stroke care and rehabilitation and wanted training from professionals on how to do this [21]. Some carers appreciated the establishment of remote forms of communication in order to stay connected [20,24,26], and the support of online self-help groups and materials which helped stimulate care recipients [24].

“*Sometimes I video call someone to let her see people; otherwise she would lose all social contacts and they are already physically distant*” (Carer [24])

Despite the efforts to stay connected, the majority of carers felt that remote support was not as good as in-person contact and that patients missed the engagement of activities and specific interventions such as cognitive stimulation therapy which had been provided at day centres [25,28,29,30].

Carers in several studies noted an increase in the dependency of the care recipient, as they were doing less tasks independently, such as shopping and socializing [19,24,25]. Carers noted that their care recipient had increased expectations of them for caregiving tasks [21], in particular if the caregiver was now working from home or unable to work [21,29]. This situation left carers feeling more exhausted, with less time for themselves.

“*He has become more demanding, if he sees me around, he would call for me more often, or just call me no matter what happens just because I am here… He might keep asking for food every 2 h… Or he might ask me to turn off the aircon, or fan, or lift him around*” (Carer [21])

Carers reported fearing that the increased caregiving burden would last beyond the pandemic [19] due to a suspected permanent decline in the person with dementia and an expectation that carers would carry out tasks such as grocery shopping that the person with dementia had done previously before the pandemic. The studies involving patients with neurological conditions [19,20,25,29] also highlighted the complications carers experienced as symptoms increased, resulting in more expectations upon the carer.


**Staying connected—“*Good enough*”**
[22]

As services closed and restrictions were enforced, carers experienced loneliness and isolation and attempted to look for new ways to stay connected through technology and ‘outside meet-ups’. Whilst the carers praised the efforts of healthcare professionals and friends and family in their attempts to provide support in new ways [20,26], they expressed concerns [27] that the temporary measures may become the new normal, which would not fully replace the in-person interaction and therapeutic settings of respite centres and day care facilities [28,29].

“*I’m just hoping all these services eventually do come back and they come back sooner rather than later because my Dad hasn’t been out*” (Carer [27])

Seven studies reported loneliness amongst the carers who struggled being isolated from family and friends during lockdown [19,20,22,24,26,28,31]. Those living with people with long-term conditions were isolating as much as possible and not allowing visitors in order to protect their care recipient from the virus, increasing feelings of loneliness. Carers of people with dementia [19] noted that social interaction with their care recipient was limited due to the care recipient’s declining cognitive status which may have been further impacted by the limited social interaction support activities. Similarly, other carers of people with dementia found the social isolation extremely difficult due to a decline in the persons health, lack of respite and subsequent impact on the carer’s wellbeing [21,22].

“*Very soon after the day care facility closed, my mother started to deteriorate in her health especially the mental one. Her behaviours started to change, a lot of problems grew and for me it was a very big problem*” (Carer [22])

For those who did not live with their care recipient, having to isolate from their family member was difficult. Again, this was emphasised in the studies involving people with dementia, where physical contact was an important way to connect and communicate with their loved one [20,22,25,27].

“*She doesn’t understand if I’m outside why I’m not going physically into the building*” (Carer [22])

Carers felt that informal care could not be replaced by a staff member in institutional settings with some breaking social distancing rules within a care facility behind a staff member’s back to hug their mother. Another carer described not being able to visit their father as “soul-crushing”, distressed by the impact that this would have on their father’s psychological wellbeing [26]. This demonstrates how many carers see in-person contact and physical affection as a key aspect of care, which they were unable to carry out during lockdown.

“*I feel deprived of just being able to sit with her and I can’t really talk to her. We would just sit there and I would just sit there and kind of rub her back and chat with her, and I am sad I can’t do that anymore.*” (Carer [20])

Carers unable to accompany people for medical appointments also reported the separation and not being able to be present as distressing and difficult in terms of missing important communication.

“*It was quite traumatising for me as well, because I wasn’t there next to her and I just didn’t know what the hell was going on*” (Carer [31])

An important way of coping during the pandemic for carers was staying connected to their care recipient if living apart, and to family and friends. Participants described using technology to video chat with their care recipient, or to keep their care recipient connected to other family members [20,22,26,29]. Technology became an important method of staying connected with loved ones for many. Other examples aside from video chatting included online support groups, social media groups, and online memory cafe groups [20,22,26,29].

“*Social media and some platforms that allow you to contact and see another person is important. It is a form that may not be perfect but it is good enough to be used*” (Carer [22])

One study highlighted that technology does not always allow for non-verbal communication to be captured which was problematic for mental health service users [28].

“*In [in-person] face-to-face meetings there was more opportunity to observe body language and assess mood from the physical presence of somebody that you’re sitting with. I don’t think that can be captured over the phone*” (Carer [28])

Others stressed the importance of finding alternative ways to visit their loved one, such as outdoor visits with social distancing measures, “drive-by visits” or “window visits” [20]. Participants stated that they felt relief when visits were allowed for recipients living in care facilities. However, they dreaded these visits being revoked in the future due to a rise in cases.

Some participants reported a positive change in the relationship with their parent receiving care [26]; the stay-at-home measures allowed more time for remote communication between family members which facilitated a greater level of openness in their relationship.

“*We’re communicating more and I’d like to think our relationships are getting closer because we’re talking about our emotions more often and more openly.*” (Carer [26])

Positive attitudes towards the lockdown were reported [18] regarding the increased amount of time carers were able to spend with their care recipients due to decreased work responsibilities or the ability to work from home.

Many carers adjusted to the altered methods of social connection and communication, with some embracing the changes [22,26]. However, several studies reported carers had difficulties with the use of technology [20,25,29]. One carer stated that their mother did not like people watching her through a screen; this study also noted barriers to the use of technology for support such as some carers did not know how to use or did not have access to technology [25]. Another recalled how their mother had no reaction to ‘face-time’ due to her dementia [31]. The usefulness of remote communication was therefore limited for some informal carers, in particular for those caring for a person with severe dementia.

“*My mother will not cooperate as she is not comfortable with watching people speaking on a screen*” (Carer [25])

Similar concerns were evident in carers of people with mental health illness,

“*I’m just fearful that [service] will take from this, ‘oh we can do it all on zoom, we can do it all online, we can do it all on the phone, we don’t need to actually see people’. And that would be a very negative thing for an awful lot of people. Those people would vanish quite rapidly*” (Carer [28])

## 4. Discussion

The aim of this review was to develop an understanding of the experiences of informal carers during the COVID-19 pandemic. The findings show that the pandemic had a significant impact on carers elevated by the fear of the virus, uncertainty of new processes, the burden associated with a lack of support but increased expectations and the attempts to stay connected during periods of isolation. The findings are important in enabling lessons to be learned which can be driven through policy and into practice, focusing on better support for carers in the future and highlighting that if a similar crisis were to happen again, what effective support looks like and how it should be delivered. Caregiving is now a significant public health concern given the rapid rise in the number of informal carers and the increased financial and social costs to society, families, and citizens. The findings reported here are timely given the current crisis of health and social care systems and the need to translate research into policies and widespread practice to ultimately improve quality of life and outcomes for carers and care recipients.

A recent scoping review [33] identified that the COVID-19 pandemic had likely exacerbated psychological distress for carers and disrupted access to formal and social supports. Similarly to our systematic review, it found that whilst some informal supports such as memory cafes and smart phones helped carers connect, the restrictions to formal support and increased isolation would likely lead to a decline in carers’ psychological wellbeing and concerns about their own physical and psychological health and financial status. Whilst a scoping review is limited in that it does not account for the quality of evidence [33], our systematic review supports their findings in identifying the serious concerns for carers’ wellbeing and their request for carers to be prioritised in social care reform. The in-depth thematic analysis in our qualitative systematic review further explains the underlying drivers for the decisions carers made during the pandemic and their actions, such as the impact of fear and uncertainty and the increased burden in the carer role and responsibilities, and identifies the challenges associated with the different disease groups.

Our findings show that the closure of formal support services and restrictions on social connections was particularly challenging for carers of people with dementia [19,22,24,25,27,31]. This is supported by a German survey of 1000 carers [34], which stated that the caregiving burden increased for those caring for a person with dementia or those who were previously receiving formal support. The increased burden was attributed to negative feelings, loss of support and increased demands and expectations. In a UK study [31], 64% of carers said they had not been able to take a break during the pandemic, with 44% saying that they were at breaking point. The evidence outlined in our review suggests that there needs to be better recognition of the carer role in supporting patients and to ensure that formal services assess carer needs as well as the person being cared for.

Many carers were unprepared for the additional roles and responsibilities they needed to take on during the pandemic. A study in Hong Kong found that almost 25% of the population had an informal caregiving role (which includes caring for children) during the early stages of the pandemic [35]; 49.5% of this group did not feel adequately prepared for the additional burdens they experienced at this time. A lack of confidence in the role and the need for education and training were reported [18] where carers of people who had a stroke took up the role of rehabilitator in the absence of rehabilitation services. Research shows that informal carers that possess higher self-efficacy have a reduced risk of caregiver burnout and psychosocial distress [36]. Assessing confidence and competence within the carer role accompanied by training for increased caregiving was absent during COVID-19. Healthcare professionals would normally play an important role in carer education, in terms of preparation and support.

Carer stress caused by the financial and economic impacts of the pandemic was evident in some studies [18,24,25]. Wider research indicates that financial burden is a bigger problem than identified in this review. Multiple studies have noted financial difficulties for carers in a variety of countries such as the US [37], Australia [38] and the UK [39]. Reasons for financial stress include having to give up work to care for a care recipient, unemployment and economic uncertainty [39]. There may well be some advantages, but no studies reported on state support schemes for employees such as ‘Furlough’ in the UK. Similarly, where homeworking may have provided much greater flexibility in some countries, again, this was not mentioned in the evidence reviewed.

Carer fears about the spread of the virus affected behaviour change and in turn led to increased caregiving burden; for example, cancelling paid formal support [19,20,24]. This was not identified in the recent scoping review [33] but has been reported in some of the quantitative evidence. Fear experienced by carers was reported by carers of cancer patients in Singapore [40] with 72.8% of carers reporting being ‘very much’ or ‘extremely’ fearful of the virus. Similarly, a survey of the Spanish general population [41] found that carers reported higher levels of post-traumatic stress symptoms compared to non-carers with the same levels of COVID-19 related fear. Our review identified that carers were wary of re-integrating into society as lockdown measures eased [19], showing even as the risk of contracting the virus lessened, carers were still afraid. Our evidence identifies important gaps in support for health and social care. There is a need to consider how best to support carers moving forwards and re-engaging with wider communities.

Carers lost their own social support networks during the pandemic. Not only were support services shut down, but social distancing measures mean that opportunities for social support were severely curtailed. Only 30% of carers in Carers UK’s study reported having a support network around them [39]. Research has shown that pre-pandemic informal carers were already at risk of loneliness and social isolation [42]. Our review found that many carers utilised technology during the pandemic to maintain social connections, such as video chat with family and friends and online Zoom support groups [20,24]. This is supported by another UK study [43] that acknowledged that over time the use of a peer support intervention for carers during the pandemic allowed friendships to develop and provided an opportunity to exchange experiences, ultimately improving wellbeing.

Remote support through technology was often available to patients and carers in lieu of in-person support, such as internet memory cafes for people with dementia. However, there were mixed views on its usefulness [20]. Remote connections can also be a helpful strategy to enable relatives to say goodbye to a dying person [44] and may help mitigate distress amongst the bereaved who had no choice but to be absent at the time of death of a loved one during the pandemic. Some studies in this review highlighted challenges with the use of technology [20,24,25], such as a lack of knowledge on how to use technology, or an inability to communicate digitally with care recipients with severe dementia. The digital divide has been reported [39], with 10% of carers reporting their ability to use technology was limited due to difficulties affording equipment or internet connection. A scoping review of information communication technology (ICT) solutions for informal carers [45] identified further barriers such as technophobia and concerns about the ‘dehumanisation of care’ where ICT is used to replace the personal component of care. Carers viewed some digital services as impersonal. It is therefore recommended that developers of ICT solutions for informal carers involve older carers in the process to improve ease of usability and improve attitudes towards technology [45]. Digital support has the potential to increase access to healthcare to a wider range of carers and efforts to improve digital literacy are warranted. However, individual preferences among carers regarding the use of technology need to be considered as services develop post-pandemic, considering the challenges that some populations experience with remote communication.

### Strengths and Limitations

This is the first systematic review of qualitative evidence exploring carers during the COVID-19 pandemic. It covers a range of experiences of various caregiver groups, including those caring for older adults and people with dementia, disabilities, end-stage renal disease, cancer, and stroke survivors. It is not surprising that that more studies have focused on dementia carers, given that this caregiving group represents a large proportion of informal carers. As we have shown, a lot can be learnt from the concerns they raise. The strength of qualitative reviews is to identify the range and diversity of experiences, in this case across carers around the world, to give insights into the different ways the pandemic has affected informal carers using the existing qualitative literature.

There are some key differences between the countries where healthcare systems operate differently, and financial issues may likely be more severe for carers. In addition, the studies reviewed were conducted in different countries that will have had different severities of (and responses to) the virus, with different healthcare systems and governmental strategies, which may have resulted in different factors impacting the themes such as fear and level of support available. However, all countries included were impacted by the virus and stated that they had some form of national lockdown.

We found limited research specifically examining the experiences of carers of people approaching end of life. Whereas the key themes identified are relevant to this population; specific experiences regarding carers’ experiences of providing end-of-life care have yet to be identified.

## 5. Implications

The evidence provided here can help guide policy and inform the development of future services, considering the specific needs of informal carers at a time where they have experienced increased responsibility but reduced support. Infection levels associated with the COVID-19 virus will continue to move up and down as the disease moves into an endemic phase. Many restrictions have lifted and whilst society attempts to navigate living with COVID-19, many unpaid carers and families have continued to ‘shield’ and withdraw from society, meaning many are still experiencing lots of the issues identified in this research. The findings of this review therefore have significant implications and recommendations important and relevant for today.

This review shows that guidance and education for informal carers is paramount during the ongoing pandemic to increase levels of confidence and competence. COVID-19 has exacerbated the needs of carers, many of which were already there. Even if the level of COVID-19 reduces or even disappears the challenges to carers identified in this paper will not and will need addressing to avoid future pressures on the health and social care system. New carers would benefit from early targeted education that can be given in digital form to enable access at a pace they can manage alongside their caregiving roles.

The review emphasised the importance of staying connected in times of isolation and service closures. Communication is vital here to ensure carers are kept up to date about the status of support services, especially if the move is to hybrid or blended services. Policymakers should also consider where possible a return to in-person services for those that benefit from it such as those with cognitive impairments. Digital services will likely remain for carers that prefer the virtual format, for example, if they have time or financial constraints, and offering options to people who need care and support over different platforms is a positive move forward. A focus outside of formal services may be needed to target loneliness and social isolation from a community approach to care [46]. The model of ‘compassionate communities’, in-part inspired by community approaches to end-of-life care in Kerala, India, has the potential to support carers by empowering communities to better support each other to be prepared and enable a good end of life wherever possible [47] and preventing social isolation by facilitating social support networks.

Formal services such as hospitals, hospices and care homes can facilitate connection between carers and their care recipients and in the event of another lockdown, one ‘essential carer’ should be allowed as a key worker to visit their loved one across all settings not just care homes [48]. This is particularly important for carers of people with dementia as this review has shown that care recipients may not respond to technology such as video chatting. Whilst complex in terms of resources, considerations need to be made for digital support for carers, such as technological support possibly through volunteer led systems to increase confidence with its usage, as well as support with other issues such as internet connection and access to technology.

Only one study in this review focused on palliative care [29], and it is likely that many carers experienced a substantial increase in responsibilities, especially for those at the end of life during the pandemic. Research is being conducted by the research team to look at the impact of the pandemic on informal carers who were bereaved during the pandemic in the UK. This will offer further insights for policy and practice in a relatively unknown territory.

## 6. Conclusions

The systematic review identified multiple sources of increased burden and an urgent need for support for informal carers during the COVID-19 pandemic. Carers have been struggling to manage the increased responsibilities such as the protective measures and reduced support services available. However, more research is needed on different caregiving groups and for those at the end-of-life. Informal carers contribute greatly to supporting healthcare systems; therefore, healthcare systems and communities need to adapt and evolve as more efficient and sustainable support networks and services as the world recovers, to meet the carers’ changing needs and preserve this vital workforce.

## Figures and Tables

**Table 1 ijerph-19-13455-t001:** Key words used in the database search.

Population	Exposure	Outcome
Informal care* or Family care* or Informal caregiv* or Family caregiv* or Informal home care* or Unpaid care* or Informal caring	COVID* or COVID-19* or pandemic* or Coronavirus infection* or 2019-nCoV* or SARS-CoV-2* or covid19*	Experience* or Challenge* Perception* Lived experience* Impact* Effect* Consequence* Perspective* or Thought* or Belief* or Emotion* or View* or Wellbeing or Qualitative or Well-being or Quality of life or Stress* or Pressure* or Mental health or Burden* or Attitude*

**Table 2 ijerph-19-13455-t002:** Summary of critical appraisal.

	Research Design	Recruitment Strategy	Data Collection	Researcher ParticipantRelationship	Ethics	Data Analysis	Clear Findings	Value of Research	Overall Impression
Sousa et al. (2021) [18]	++	+	++	−	+	+	+	+	+
Hanna et al. (2021) [19]	++	+	++	−−	+	+	+	+	+
Lightfoot et al. (2021) [20]	++	+	++	−	−	+	+	+	+
Lee et al. (2021) [21]	++	+	++	−−	+	+	+	+	+
Mackowiak et al. (2021) [22]	++	++	++	−−	−	+	++	+	+
Chia et al. (2021) [23]	++	−	++	−−	+	++	+	+	+
Cipolletta, Morandini and Tomaino (2021) [24]	++	+	++	−−	+	+	+	+	+
Vaitheswaran et al. (2020) [25]	++	++	++	−−	+	−−	+	+	+
Fisher et al. (2021) [26]	++	+	++	−−	+	++	+	++	+
Giebel et al. (2021) [27]	++	++	++	−−	+	++	+	+	++
Liberati et al. (2021) [28]	++	+	+	−−	+	+	+	+	+
Macchi et al. (2021) [29]	++	++	++	−−	++	++	++	++	++
Simblett et al. (2021) [30]	++	+	+	−−	++	+	+	++	+
West et al. (2021) [31]	++	+	+	−−	+	+	++	++	+

Key: ++ = Good; + = Fair; −− = Poor.

**Table 3 ijerph-19-13455-t003:** Summary of study characteristics.

Author, Country and Long-Term Condition	Aim	Methods	Key Findings
Sousa et al. (2021) [18] Portugal Renal Failure	To explore relatives experience of caring for people with end stage renal disease during COVID-19.	Semi-structured telephone interviews with family carers (*n* = 19) Thematic analysis	Carers experienced emotional distress Changes in caregiving responsibilities Educational and supportive needs Coping strategies to deal with lockdown
Hanna et al. (2021) [19] UK Dementia	To explore the impact of COVID on people living with dementia and their unpaid carers	Semi-structured telephone interviews Convenience sample of carers (*n* = 42) and people with dementia (*n* = 8) Thematic analysis	Communication (strengthened resilience offered security and enabled burden to be expressed) Adaptions (technology, PPE) Support networks (family, community support, support groups online, paid homecare) Lifestyle factors and coping mechanisms (‘green space’)
Lightfoot et al. (2021) [20] US Older adults and Learning Disabilities	To explore family carers views of how caregiving for older adults and people with disabilities has changed during COVID-19	Semi-structured video interviews with carers (*n* = 52) of older people or people with learning disabilities Convenience sample (*n* = 52) Thematic analysis	Limited social and physical contacts (no in-person contact, reduced networks) Changed caregiving tasks (less social stimulation, changed schedules, more practical caregiving) Reduced services (less formal support) Focus on vigilance and safety (PPE, keeping safe) New ways of connecting and getting support
Lee et al. (2021) [21] Hong Kong Stroke/CVA	To explore the experiences of carers of stroke patients during the pandemic	Semi-structured telephone interviews with carers (*n* = 25) Purposive Sample Thematic analysis	Limited care services Increased workload and strain Threatened relationships between caregiver and stroke survivor Threats to carers well-being (increased care burden and reduced support)
Mackowiak et al. (2021) [22] Poland Dementia	To explore the experiences of people with dementia and informal carers during COVID-19	Semi-structured interviews with people with dementia (*n* = 5) and their carers (*n* = 21) Convenience sample Thematic analysis	Care reorganisation (restricted facilities or shut down of services, access to medical help and burden of care) Psychological response of uncertainty and anxiety, social isolation and adaption and coping. Emerging needs of institutional support, social support and remote contacts
Chia et al. (2021) [23] Singapore Cancer	To explore the emotional impact and behavioural response to COVID amongst cancer patients and their carers	Semi-structured interviews with patients (*n* = 16) and carers (*n* = 14) Purposive sample Thematic analysis	Heightened sense of threat (fear and vulnerability, uncertainty) Necessary disruptions by new procedures Increased responsibility Strive for ‘normalcy’ Safety and Trust
Ciopolletta et al. (2021) [24] Italy Dementia	To explore the experiences of family carers of people with dementia during the COVID-19 pandemic and the impact on their lives	Semi-structured interviews with carers (*n* = 20) Convenience sample Thematic analysis	Practical difficulties with everyday care and time needed in care routines Emotional stress Difficulties reaching out for help
Vaitheswaran et al. (2020) [25] India Dementia	To explore the experiences of carers of people with dementia during COVID-19	Semi-structured telephone interviews with carers (*n* = 31) Purposive sample Thematic analysis	Multiple needs requiring pragmatic multi-layered approach Changes to daily routines Managing behavioural psychological symptoms challenging
Fisher et al. (2021) [26] US Cancer	Explore how COVID-19 affected adult children caring for parents with blood cancer	Survey (*n* = 84) carers with open questions for thematic analysis Convenience sample Thematic analysis of open responses	Increased fears and uncertainty-related stress Reduced in-person care opportunities Increased isolation Enhanced family communication
Giebel et al. (2021) [27] UK Dementia	To explore perceptions of public health restrictions for carers of people with dementia during the pandemic	Telephone interviews with carers (*n* = 50 and *n* = 20 follow up) Convenience sample Thematic analysis	Confusion and limited comprehension Frustration and burden Putting oneself in danger Adherence to restrictions in wider society Unchanged perceptions
Liberati et al. (2021) [28] UK Mental Health	To explore experiences of remote care for mental health care users and carers during COVID- 19.	Interviews (*n* = 65) (Telephone or online) with carers (*n* = 10), service users (*n* = 20) and staff (*n* = 35) Convenience sample Thematic analysis	Therapeutic encounters (social isolation, loss of therapeutic setting and quality of care) Remote assessments and identifying risks Inequities in access Future of remote care (less shared decision making and technical issues)
Macchi et al. (2021) [29] US Neuro-Palliative Care (Dementia/Parkinsons)	To explore patient and caregiver perspectives of neuro-palliative care during COVID-19	Qualitative arm to RCT Semi-structured interviews (teleconference/phone) with patients with Parkinson’s or Alzheimer’s disease (*n* = 108) and carers *n* = 90) and medical documentation Randomised sampling using stepped wedge approach Thematic analysis	Disruptions to healthcare and support services Increased symptomatic and psychosocial needs Increased caregiver burden Limitations of telecommunications compared to in- person contact
Simblett et al. (2021) [30] UK Mental Health	To explore perspectives of mental health service users and carers during COVID-19	Interviews with service users (*n* = 18) carers (*n* = 5) and (*n* = 8) identified as both. Convenience sample Thematic analysis	Emotional responses (Fear of infection, death and future; anger, sadness, guilt, boredom, happiness/calm at slower pace of life) Thoughts about life situation, relationships and services Behaviours (technology, impact on social behaviours/leisure activities)
West et al. (2021) [31] UK Dementia	To explore the impact of COVID-19 on people with dementia and carers from black and minority ethnic groups	Interviews with carers (*n* = 11) and people with dementia (*n* = 4) Purposive sampling Thematic analysis	Fear and anxiety Lifestyle changes (shopping, eating patterns, Isolation and identity Community and social relationships Adapting to restrictions and new ways of living Social isolation and support structures Medical interactions Planning and decision making

## Data Availability

Not applicable.

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
