# Peer review of "The Experiences of Informal Carers during the COVID-19 Pandemic: A Qualitative Systematic Review"

_ijerph, 2022, doi:10.3390/ijerph192013455_

Round 1
Reviewer 1 Report
It is an interesting, novel study with enormous social repercussions. It may help health and social-health care service managers to develop and implement strategies that help informal caregivers to assume responsibility for the sick family member. One of the most relevant conclusions is that it shows the importance of educating caregivers to be able to maintain the sick person at home. This education should be done as soon as possible. They also suggest other interesting options such as a return to face-to-face care and adopting a strategy to mitigate isolation, such as the "compassionate communities" model.
However, the authors should improve some aspects mentioned below.
Methodological aspects:
If authors want to analyze the scientific evidence of qualitative design articles, they should not use PRISMA but COREQ, or the list proposed by the Joanna Briggs Institute. The following links are recommended.
https://jbi.global/critical-appraisal-tools
https://www.equator-network.org/reporting-guidelines/coreq/
They should include the meaning of the acronyms introduced in the text (In the Quality appraisal section there are acronyms not defined elsewhere) Who are RT and CB?
Regarding tables and figures:
· * Figures must have correlative numbering according to appearance in the text, Figure 2 should not be before Figure 1. Where is Figure 1?
· * Is Figure 2 what is called a "mental map"? where is it?
· * Table 3 is not cited in the text.
Specific comments
· Theoretical foundation: It is brief but focused on the topic.
· Research question: Table 1 containing the key words used in the search following the parts of the research question is very good.
· Inclusion and exclusion criteria: Correct
· Adequate quality assessment tool of the studies that make up the Unit of Analysis (14 papers were suitable for the review (CASP)). The method of data extraction and data analysis is adequately detailed.
· The authors say that they have used a tool to detect duplicates, could they cite it?
· Results: Describes very well the critical appraisal of the selected articles, and their characteristics. Excellent description of the content of each identified topic.
· Discussion: Very well elaborated since it contrasts with other studies well related to the topic, and which are not part of the Unit of Analysis of the present study.
· Limitations. They explain well the strengths and weaknesses of the study. They also identify new lines of research: "exploring caregivers' end-of-life experiences".
· In the implications section, the authors state that only one study focuses on caregivers at the end of life, could you please insert the citation?
· Conclusions: they follow from the analysis and are concrete.
· Bibliography: in the bibliography section there are incorrect and incomplete citations. There are some citations not inserted in the text that later appear in the bibliography. This section should be worked on thoroughly.
Author Response
Thank you for this review and for your detailed comments on our paper. Firstly our apologies for the errors with the figure and table numbers and for submitting the incomplete reference list. Thank you for highlighting these and giving us the opportunity to amend our oversight. We hope the following response and corrections improve the manuscript to your satisfaction:
- The reason we chose PRISMA to report the paper was due to it bing a systematic review, we felt this was more appropriate and offered more detail in the report than the COREQ reporting criteria for qualitative research.
- RT and CB are authors in the paper - clarification of RT added to contributorship statement at the end
- Figure numbers have been clarified - there is no figure 2, thank you. Table 3 is citied in the text on page 10.
- Clarification over removing duplicates given.
- Reference added to palliative care focused study in discussion as suggested - thank you.
- Correct reference list inserted, many apologies.
Thank you for your feedback and suggestions. Please do let us know if you require anything further.
Reviewer 2 Report
The paper adds a valuable insight into the impact of the pandemic upon informal carers. A good addition to this developing topic area. The paper is well written and methodologically sound. Themes, as expected, are supported by the participants' voices (quotes).
Author Response
Thank you very much for your positive review.
Reviewer 3 Report
Overall, this paper is great. It adds a lot to the field of public health. Why not just call them informal caregivers as that is the consistent term used within the literature? When talking about covid it should be COVID-19 as this is the name given for the Coronavirus pandemic.
Some of the discussion is lacking regarding what these findings mean? In other words, why should we care? After reading this section I thought, why are these findings important?
Author Response
Thank you very much for this positive review. You raise an important point about the term 'informal caregivers' and one we as a research term have discussed at length. Based on your recommendation, on page 4 we have added a justification to show that whilst we recognise the variation in terms we will use 'informal carers' for consistency in reporting the findings. However in the table of articles we have remained true to the terms the original papers used.
Thank you for suggesting we strengthen the opening of the discussion. On page 25 we have aded:
The findings are important in enabling lessons to be learned which can be driven through policy and into practice focusing on better support for carers in the future and highlighting that if a similar crisis were to happen again, what effective support looks like and how it should be delivered. The findings are timely given the current crisis of health and social care systems.
Many thanks for your review and feedback. If you require anything further, please do let us know.